# Three Quarters of a Century of Research on RF Exposure Assessment and Dosimetry—What Have We Learned?

**DOI:** 10.3390/ijerph19042067

**Published:** 2022-02-12

**Authors:** Kenneth R. Foster, Marvin C. Ziskin, Quirino Balzano

**Affiliations:** 1Department of Bioengineering, University of Pennsylvania, Philadelphia, PA 19104, USA; 2Department of Radiology, Temple University Medical School, Philadelphia, PA 19140, USA; ziskin@temple.edu; 3Department of Electrical and Computer Engineering, University of Maryland, College Park, MD 20742, USA; qbalzano@umd.edu

**Keywords:** radiofrequency radiation, exposure assessment, dosimetry, history, exposure limits

## Abstract

This commentary, by three authors with an aggregate experience of more than a century in technology and health and safety studies concerning radiofrequency (RF) energy, asks what has been learned over the past 75 years of research on radiofrequency and health, focusing on technologies for exposure assessment and dosimetry. Research programs on health and safety of RF exposure began in the 1950s, initially motivated by occupational health concerns for military personnel, and later to address public concerns about exposures to RF energy from environmental sources and near-field exposures from RF transmitting devices such as mobile phones that are used near the body. While this research largely focused on the biological effects of RF energy, it also led to important improvements in exposure assessment and dosimetry. This work in the aggregate has made RF energy one of the best studied potential technological hazards and represents a productive response by large numbers of scientists and engineers, working in many countries and supported by diverse funding agencies, to the ever rapidly evolving uses of the electromagnetic spectrum. This review comments on present needs of the field, which include raising the quality of dosimetry in many RF bioeffects studies and developing improved exposure/dosimetric techniques for the higher microwave frequencies to be used by forthcoming communications technologies. At present, however, the major uncertainties in dosimetric modeling/exposure assessment are likely to be related to the inherent variability in real-world exposures, rather than imprecision in measurement technologies.

## 1. Introduction

This Special Issue of IJERPH concerns human exposure to radiofrequency (RF) energy, both in occupational and nonoccupational settings. The present authors have an aggregate experience of more than a century in RF technology and health and safety studies concerning this form of nonionizing radiation and offer these brief historical comments. This commentary provides a brief overview of the impressive advances in RF dosimetry and exposure assessment over the past three quarters of a century and outlines some current issues in integrating technical advances in these areas with scientific studies to improve the understanding and control of the possible health and safety issues with RF energy. 

In the following discussion, ‘exposure assessment’ refers to evaluation of levels of RF energy incident on the body, and ‘dosimetry’ refers to determining the absorption of RF energy within the body. Technical reviews of these topics are found in [1,2] and other sources. RF bioeffects, either thermal or nonthermal in mechanism, are outside the scope of this commentary. References were chosen to illustrate major advances or provide background information; this is not a systematic or fully comprehensive review, a project that would be out of scope of the present commentary.

## 2. Origins in Medicine

A century ago (1920) the first commercial radio station, KDKA, began broadcasting in Pittsburgh, sparking a revolution in the dissemination of information and entertainment directly to the public via radiofrequency signals [3]. The need to assess human exposure to RF energy gained importance with the advent of RF diathermy, which was first developed commercially by the General Electric Company (Schenectady, NY, USA) and Siemens (Erlangen, Germany) in the 1920s and subsequently widely adopted in physical medicine for therapeutic heating of tissue [4]. 

Early pioneers in bioelectromagnetics, including Herman Schwan (1915–2005) and Arthur W. Guy (1928–2014), began their research careers investigating therapeutic uses of RF energy. Schwan did his doctoral, and later habilitation, research on electrical properties of tissue at radiofrequencies under the prominent radiation biologist Boris Rajewski (1893–1974), who had become interested in therapeutic applications of RF energy. After the war, Schwan was brought to the U.S. by the Navy and established a laboratory at the U.S. Naval Base in Philadelphia (1947). In 1950 he took an appointment in the Department of Physical Medicine at the University of Pennsylvania. He soon (1952) received a secondary appointment in the engineering school, where he remained for the remainder of his career [5]. 

Originally trained as a radar engineer, Guy began his academic career in the Department of Physical Medicine and Rehabilitation at the University of Washington, which was led by Justus Lehmann (1921–2006), a distinguished physiatrist with special interest in diathermy. Guy maintained that appointment for many years, but took an adjunct appointment in the electrical engineering department as well as, at the university’s Center for Bioengineering. As head of the Bioelectromagnetics Research Laboratory, he maintained a longstanding research program on the bioeffects of RF energy. 

Occupational safety concerns became a growing motivation for RF bioeffects studies. By the early 1950s, the US Department of Defense (DoD) had in operation a large number of high-power RF transmitters including communications and radar systems, with many shipboard transmitters operating in close proximity to personnel. Concerned about possible occupational hazards of such systems, DoD set up the Tri-Services Program (1956–1960), which funded RF bioeffects studies by prominent investigators at ten academic centers, including Schwan’s laboratory. The studies were small but overall were carefully done, with due attention to dosimetry within the capabilities available at the time. 

However, as one review put it, ‘the art of microwave dosimetry was just emerging’ [6]. Methods available at the time could provide only limited information about the distribution of absorbed energy in the exposed animal, and temperature control was often a problem. Schwan’s research focused on biophysical mechanisms of the interaction of RF fields with tissue and the dielectric properties of biological materials. Guy and his colleagues pursued more applied topics. Starting in the early 1970s, he and his student, C-K Chou (now a senior leader in bioelectromagnetics and RF standards setting) carried out a series of animal studies that, even by present-day standards, stand out for high quality exposure assessment. Their technical advances include thermographic imaging of split, scaled-down models of the human body [7] to determine patterns of RF absorption within the body. Chou was apparently the first to use the term specific absorption rate (SAR) in his PhD. thesis in 1975 [8].

## 3. Controversies

The publication in 1962 of Rachel Carson’s Silent Spring raised the public’s awareness of possible environmental and health risks of technologies of all sorts. In 1976 environmental journalist Paul Brodeur published two tendentious articles in the *New Yorker* about the health risks of RF energy, which were collected in his subsequent book *The Zapping of America* (1977) [9]. Brodeur presented a litany of complaints about harms from occupational and nonoccupational exposures to RF fields, at levels far below those resulting in immediate injury. Brodeur also called attention to the much lower RF exposure limits in the Soviet Union vs. those in the U.S. and other Western countries (a similar difference still exists today [10]). 

In part because of health concerns, citizens’ protests built against major projects, for example a proposed radar system (PAVE PAWS) in Cape Cod MA. In response, the MA Dept. of Public Health appointed an expert panel (that included one of the present authors). The panel assessed the bioeffects and epidemiology studies available at the time. ‘There are a number of publications that report biological effects after exposure to [RF power densities] at levels lower than the ‘safe’ limits given in national standards’, the panel noted, but ‘the evidence for these ‘low level’ effects does not reach a level sufficient to justify claims of any health hazard.’ [11]. The report (which was released in final form in 1999) called for surveys of RF exposure to the population near the proposed facility as well as more research.

As new RF technologies were introduced, they also became controversial due to safety concerns: microwave ovens (1970–80s), police radar (1980s), mobile phone handsets (starting in the 1990s; one of the present authors designed the antennas used in the first commercial mobile phones by Motorola), Wi-Fi, wireless-enabled utility meters (“smart meters’) and, at present, 5G cellular systems. In addition to many laboratory studies reporting biological effects from exposure to RF energy at widely varying exposure levels, the epidemiology literature now includes numerous reports of associations between occupational and nonoccupational exposures of some sort to RF energy and diverse health endpoints, including several forms of cancer.

## 4. Internationalization of Bioelectromagnetics Research

As a result of these many controversies, research on RF health and safety evolved, from research focused on obvious hazardous effects from relatively high exposure levels to address occupational safety concerns, chiefly done in the U.S. under military support, to much more diverse research in many countries, initially in Europe but later Asia and Oceania (Australia), and often in search of effects at levels far below national guidelines. These scientific studies have gone hand-in-hand with technical studies to improve dosimetry and exposure assessment. Major advances in dosimetry and exposure assessment over the years include the following:

### 4.1. Rise of Computational Dosimetry

Starting in the 1970s, theoretical dosimetry took on a life of its own, apart from its use in analysis of particular experiments. Kritikos and Schwan at the University of Pennsylvania (e.g., [12]) and Durney and colleagues at the Univ. of Utah [13] among others, obtained analytical solutions to the electromagnetic field equations for simplified geometrical models of the body exposed to plane wave RF radiation. Other investigators, notably Gandhi and colleagues at the University of Utah (e.g., [14]), developed approximate computational techniques that initially used crude block models of the body but later highly realistic numerical models based on medical imaging. This early work was summarized in a series of dosimetry handbooks sponsored by the U.S. Air Force beginning in 1976 [15], which are still useful resources and available online. Much of this early work considered far-field exposure to plane wave radiation, perhaps reflecting early DoD interest in occupational exposures from RF transmitters and perhaps also because of the more challenging problem of assessing near-field vs. far-field exposures. 

Due to the limited range of idealized models that are amenable to exact analytical solutions, and enabled by the steady increase in computing power, numerical modeling came to supplant analytical studies. The finite-difference-time-domain (FDTD) method, which was introduced to bioelectromagnetics in 1987 by Gandhi’s group (Univ. of Utah) [16] and is well adapted to RF field calculations above ≈100 MHz, and has become the de facto standard approach to numerical dosimetry. This work was facilitated by the contributions of Gabriel and colleagues, who produced a very extensive survey of the dielectric properties of numerous human and animal tissues between 10 Hz and 20 GHz [17]. With approximately 4500 cites (according to Google Scholar) as of early 2021, this is perhaps the most-cited paper in bioelectromagnetics.

Presently, more than 500 dosimetric studies have been published using the FDTD method. Highly polished commercial FDTD programs exist, including very extensive packages of electromagnetic and thermal modeling programs specialized for bioelectromagnetics research by Zurich Med Tech and SPEAG (both in Zurich). ‘Off the shelf’ numerical models of the body based on high-resolution medical images are presently available, which allow assessments of the distribution of absorbed power in the body (specific absorption rate below 6 GHz and epithelial power density at higher frequencies), as well as predictions of the resulting temperature rise in the exposed tissue.

### 4.2. Comprehensive Assessments of Environmental Exposures

In the mid-1970s, Tell and Mantiply carried out the first comprehensive survey of environmental RF fields in urban environments for the U.S. Environmental Protection Agency (EPA) [18]. This survey showed that the strongest environmental RF fields in U.S. urban areas were produced by FM radio and VHF television broadcasting stations; exposure levels in publicly accessible spaces were far below the exposure limits then in effect. However, other measurements by Tell showed that RF exposures close to high-powered FM broadcast station antennas could easily exceed general public or occupational exposure limits [19]. The U.S. Federal Communications Commission (FCC) adopted its first exposure limits in 1985, and later produced a series of technical bulletins providing guidance for implementing the Commission’s exposure limits, focusing on exposures from broadcast transmitters and other high power sources that were of regulatory concern. 

Following the increased public concern about possible health effects from environmental RF sources, numerous surveys have measured environmental RF fields from diverse technologies including broadcast transmitters, cellular base stations, mobile phones and other communications sources in environments as diverse as schools, workplaces, homes, trains and other vehicles (e.g., [20]). Recently, van Wel et al. carried this farther, estimating organ-specific exposures from environmental exposures to RF fields [21]. These studies together present a detailed picture of environmental RF exposures, nearly all of which are at levels far below internationally accepted safety limits. 

These surveys were enabled by advances in instrumentation for exposure assessment over the years. The EPA-sponsored survey in the mid-1970s employed a van filled with calibrated antennas and spectrum analyzers that the investigators drove around the country and set up in chosen locations. Since then, portable instrumentation has been developed to allow frequency-selective measurements, for example the Narda SRM 3000 selective radiation meter (first referenced in the RF bioeffects literature in 2010) that combines a three-axis electric field probe and spectrum analyzer in an accurate and sensitive (but quite expensive) handheld instrument. This frequency selectivity enables the user to distinguish the source of a measured signal, and in advanced instrumentation, such as the Narda SRM 3000. even down to the particular wireless channel utilized by a cellular service or Wi-Fi network.

Broadband handheld RF meters have been available for many years, chiefly intended for use in occupational settings for worker protection, and remain widely used for that purpose. More recently, a variety of inexpensive RF exposure meters have appeared on the consumer market that allow consumers to measure RF fields in their homes to address their health concerns, and, apparently for other purposes as well. For example, Amazon.com lists dozens of inexpensive RF meters; one unit (the Erickhill EMF meter, USD 39) is sold as a ‘great tester for home EMF inspections’ as well as for ‘ghost hunting’. Such meters have very high sensitivity (needed to measure environmental RF fields at levels that would be found in most nonoccupational environments). Some have RF pulse detectors of very high sensitivity and emit ‘clicks’ reminiscent of Geiger counters in the presence of signals from Wi-Fi and other digital networks. Due to their high sensitivity and uncertain accuracy [22] they are unsuitable for occupational safety evaluations. However, EMF consultants employ such meters to assess RF exposures in consumers’ homes and provide mitigation services even though the exposure levels may be far below internationally accepted limits (https://buildingbiologyinstitute.org/find-an-expert/certified-consultants/building-biology-environmental-consultants/) (accessed on 17 January 2022).

Personal dosimeters, a third class of exposure assessment devices, are small body-mounted devices that sample environmental fields, classify them into frequency bands, and store data for later readout. Early devices were crude and inaccurate; one 1970 evaluation [23] considered a pocket RF dosimeter be ‘totally unsuitable as a quantitative instrument’, due to poor accuracy. By the late 2000’s personal exposimeters had evolved into refined instruments capable of monitoring an individual’s exposure to multiple RF sources over periods of a day or more, with sufficient accuracy to be useful for epidemiology studies [24]. 

### 4.3. Development of Methods to Assess Human Exposure to RF Radiation from Mobile Phones 

In 1996 the FCC adopted its first limits on SAR produced in the body by ‘portable’ RF emitting devices (which are intended to be used close to the body) [25]. This created the need for accurate and reproducible methods for SAR testing, which was addressed by the development of international standards for measurement of near-field exposures to mobile phones (IEEE 1528 series) and to hand-held and body-mounted wireless communications devices (IEC 62209), as well as by the development of sophisticated SAR testing equipment by IT’IS (Zurich). The International Electrotechnical Commission (IEC) has developed additional standards for RF exposure assessment, for example IEC 62232 for assessment of RF exposure from cellular base stations.

### 4.4. Incorporation of Improved Dosimetry in Exposure Limits

The first RF exposure limit in the U.S., USAS C95.1-1966, was developed by a committee chaired by Herman Schwan. The standard had a single frequency-independent limit: an incident power density of 100 W/m^2^ for 10 MHz–100 GHz averaged over 6 min. The limit applied for both whole or partial body exposure. The limits were based on a review of the scientific literature, as well as on approximate paper-and-pencil calculations of tissue heating by RF fields. This standard was later updated and revised under the auspices of the Institute of Electrical and Electronics Engineers (IEEE), with subsequent editions building on improvements in exposure assessment and dosimetry, as well as on updated reviews of the bioeffects literature. 

The following comments pertain to the IEEE family of exposure standards, denoted generally as IEEE Standard C95.1- x. For a comparative review of RF exposure limits see [26] and for description of the latest IEEE C95.1-2019 standard see [27]; for a detailed historical review of the development of U.S. exposure limits see Chapter 12 in [28]. The International Commission on Nonionizing Radiation Protection (ICNIRP), an independent international organization, introduced generally similar guidelines. For a history of the development of ICNIRP guidelines see [29].

Confusingly, the two sets of limits employ different, and sometimes changing, terminology. IEEE refers to its limits as ‘standards’, while ICNIRP uses ‘guidelines’. ICNIRP terms the maximum permissible exposure measured outside the body as the ‘reference level’, and the maximum permissible dose (SAR) inside the body as the ‘basic restriction’. IEEE has used different terminologies for these quantities in different editions of its standard. 

The IEEE C95.1 standard has been revised and updated several times since its first (1966) edition. Some of the more important revisions include:

C95.1-1982 (300 kHz–100 GHz) specified frequency-dependent limits, in terms of field strength or incident power density. The limits (equivalent to reference levels in ICNIRP) had a ‘U’ shaped dependence on frequency, with a broad minimum between 30–300 MHz reflecting the frequency-dependence of RF absorption in the body of a human standing erect in a vertically polarized RF field, as established by Durney, Gandhi, and others. This standard introduced a limit for ‘spatial peak’ SAR of 8 W/kg (equivalent to ICNIRP’s basic restriction) over one gram of tissue. The concept of ‘spatial peak’ SAR was incorporated into the 1996 FCC exposure limits, leading to the “SAR testing” requirements for cell phones used throughout the world. All C95.1 standards from 1991 to the present distinguish between two tiers of exposure, for exposures in “controlled” and “uncontrolled” environments, which correspond to occupational and general public limits in ICNIRP.

C95.1-2005 was based on what the standards document described as a ‘complete reassessment of the technical rationale’ and was designed to protect against “‘scientifically established adverse health effects in human beings resulting from exposure to radio frequency electromagnetic fields’. The documentation included a detailed analysis of adverse effects in animals of potential relevance to humans. In addition, this edition made several adjustments to the limits based on dosimetric considerations.

IEEE C95.1-2019, the most recent edition, incorporated numerous changes, chiefly at frequencies above 6 GHz, to address the shallow energy penetration depths in body tissues at those high frequencies. For frequencies above 6 GHz, IEEE C95.1-2019 introduced a new dosimetric concept, ‘epithelial power density’ (‘absorbed power density’ in ICNIRP). Notwithstanding differences in terminology, the two sets of limits are now largely “harmonized”, i.e., brought into agreement.

As a result of these developments, exposure limits have evolved that rely increasingly on dosimetry and exposure assessment, although without major changes in the underlying scientific rationale. Perhaps inevitably, the limits have also become far more complex with increasingly technical documentation. Assessing compliance with reference levels (exposure assessment) is relatively straightforward and can be done with ordinary RF survey equipment. However, assessing compliance with the basic restrictions (dosimetry) requires determination of absorbed power within the body, which is a research-level task needing specialized instrumentation, computational expertise, and a carefully specified methodology. 

## 5. What Have We Learned?

The development of technologies for exposure assessment and dosimetry comes hand in hand with scientific developments in bioelectromagnetics. Figure 1 (compiled from a database of publications at EMF-portal.org) compares the number of papers published per year on technical issues including dosimetry and exposure assessment, with experimental and epidemiology studies related to RF exposure of some sort. In recent years, approximately 120–140 experimental studies, 40 dosimetry/exposure assessment studies, and 20 epidemiology studies have been published per year, making radiofrequency energy one of the best-studied physical agents with respect to occupational or environmental exposures. Technical advances in dosimetry/exposure assessment have contributed immensely to the success of this field.

In brief, we have learned a great deal since the days of the Tri-Services Program. Major technical advances in exposure assessment and dosimetry include:

Computer programs and associated libraries of tissue electrical and thermal properties are available off-the-shelf that allow high-spatial resolution calculations of absorbed RF energy in the body and modeling of the resulting temperature increase.

Libraries of image-based models of human and animal bodies are available from commercial sources (e.g., IT’IS Virtual Family series of human models) or custom-built by individual investigators. These enable detailed dosimetric studies on individuals of different gender, age, and race.

High-quality instruments are commercially available for RF exposure assessment. At the high end are spectrum-analyzer based instruments that provide frequency-selective measurements. These enable the user to identify particular sources of exposure and apply frequency-dependent exposure limits. Unfortunately, such equipment is quite expensive and requires specialized expertise to operate properly, particularly for low duty cycle pulsed fields. Broadband RF survey meters are much less expensive and easier to operate.

Personal RF exposimeters are available (e.g., ExpoM-RF (Fields at Work, Zurich) and EmeSpy 200 (Microwave Vision Group, Paris)). These record RF fields in multiple frequency bands and can store exposure data over extended time periods [30], and are useful for epidemiology studies, to provide approximate records of exposure. 

## 6. What Needs to Be Done?

The fruits of many years of technical development have included huge improvements in dosimetry/exposure assessment since the days of the Tri-Services program in the 1950s. Undoubtedly, the uncertainties due to technical limitations in the exposure assessment conducted according to current best practices will be less than the inherent variability of human exposure. However, as bioeffects research increasingly turns to higher radiofrequencies, more work is needed to improve the ‘match’ between the needs of scientific research or other needs and the capabilities of the available dosimetric/exposure assessment tools. Examples include:

### 6.1. Improve Dosimetry in Biological Studies

Simkó et al. [31], Vijayalaxmi and Prihoda [32], and others have noted the high risk of bias in many RF bioeffects studies, with inadequate dosimetry being one of several frequently encountered problems that compromise the validity of the studies. Indeed, one still sees papers in which the RF exposure source consisted of a mobile phone handset placed near the experimental preparation, with no control of the phone’s output or assessment of the SAR in the preparation. Poorly done studies provide no useful scientific information, but are easily taken up in public debates about the safety of RF energy. 

The very high cost of equipment and adequate engineering support required for accurate dosimetry is a formidable obstacle to new groups seeking to enter the field. This is particularly true for studies involving millimeter waves (30–300 GHz), where both dosimetry and adequate temperature control are more challenging than at lower frequencies, due to the shallow energy penetration depth in tissue at these higher frequencies (e.g., <0.5 mm skin depth at 30 GHz).

### 6.2. Improved Techniques for Exposure Assessment/Compliance Testing for High Frequency Communications Technologies

The 5G New Radio (5G NR) cellular technology utilizes higher RF frequencies than preceding generations of cellular technology, approaching or entering the mm-wave band above 30 GHz. In addition, 5G NR technology employs MIMO (multiple input, multiple output) antennas that direct independently steerable beams that can follow the movements of individual subscribers. Dosimetry and exposure assessment of mm-waves will require approaches that are not currently used for lower-frequency cellular technologies.

This problem has two parts:

#### 6.2.1. Assessing Environmental Exposures from MIMO Antennas

Exposure assessment for transmitters employing MIMO antennas is more complex than for conventional antennas, whose beams are fixed in space. One approach is to use statistical methods to calculate distributions of exposure, and this is currently being studied by IEC/IEEE ad hoc committees, as well as equipment manufacturers. However, statistical exposure assessment is likely to employ proprietary software and will be inherently less transparent than exposure calculations for fixed-beam antennas, for which beam patterns are publicly available and calculation methods are specified by the FCC. 

#### 6.2.2. Improving Dosimetry for Near-Field Exposures at Frequencies > 6 GHz

Determining the absorption of RF energy in the skin from sources close to the body at frequencies > 6 GHz is complicated by its shallow penetration depth into tissue. At these higher frequencies, the incident power density at the skin surface, as inferred from the antenna output power and directivity, yields an inaccurate measure of the absorbed energy in tissue. This will require the development and validation of new methods for compliance assessment for handsets to be used near or against the body, a problem that remains under investigation [33,34,35].

### 6.3. Accounting for Inter- and Intrasubject Variability

Regulatory limits are expressed in terms of absorbed power in tissue, which as a practical matter is estimated by calculations or measurements on phantom materials used to simulate tissue. However, the electrical properties of tissues near the surface of the body, and hence absorption of RF energy, change due to variations in skin blood flow and moisture content, the thickness of skin and subcutaneous tissue layers, and other factors. More work is needed to establish procedures to assess the compliance of mm-wave devices used against the body because of the high anticipated variability in the small volumes of exposed tissues at such frequencies. 

## 7. Conclusions

In summary, seven decades of research on dosimetry/exposure assessment have led to the development of technologies that permit very detailed and accurate evaluation of RF exposure to the human body, both in the near field of devices such as mobile phones, and in the far field from environmental sources. However, accurate exposure assessment and dosimetry can require expensive equipment and specialized engineering support and is not always achieved in bioeffects studies.

Progress on dosimetry/exposure assessment is continuing at the higher microwave frequencies that used by 5G NR systems and other mm-wave communications systems. Based on the previous success of this endeavor, we can expect that the tools to assess the exposure of biological systems to mm-wave radiation from 5G NR and other applications of this part of the RF spectrum will, in time, reach the level of sophistication and accuracy that is currently achieved for RF signals at lower frequencies. 

## Figures and Tables

**Figure 1 ijerph-19-02067-f001:**
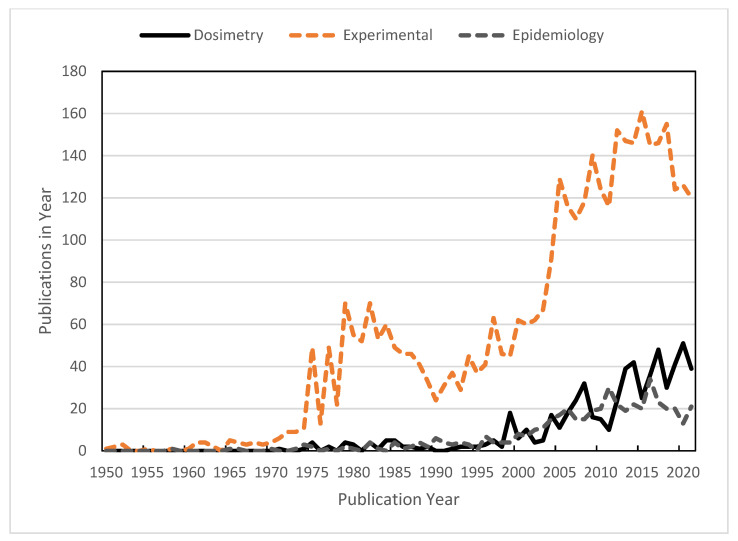
Papers published per year related to bioeffects of RF fields: technical studies including dosimetry and exposure assessment (615 studies total), experimental studies (3800 papers total), and epidemiology studies (472 papers total). Compiled from the database of publications at emf-portal.org.

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
