# Peer review of "Three Quarters of a Century of Research on RF Exposure Assessment and Dosimetry—What Have We Learned?"

_ijerph, 2022, doi:10.3390/ijerph19042067_

Round 1
Reviewer 1 Report
This work is a very clear and interesting commentary on the most important milestones reached in RF exposure assessment and dosimetry.
The paper is well organized and adequately comments the principal outcomes of the researches done up to now and on the challenges to be addressed by next forthcoming studies.
As this is a commentary and not a review paper, it is reasonable that the reference section reports only a limited number of citations. However, the authors should consider to add a few lines to explain what was the rationale they followed to choose the references they put in the paper, obviously not for what concerns references to standards or recommendations (for which they correctly reported all the relevant papers) but for the references to past research studies.
Author Response
We added a line in the introduction explaining the rationale for choosing references
Reviewer 2 Report
The paper is well written and of interest to the reader.I have only one thing that needs attention; in Fig 1 it needs to be explained what the different symbols represent.
Author Response
The figure has been redrawn and caption made clearer
Reviewer 3 Report
The article presents a very well documented review on the evolution of RF exposimetry and dosimetry field. The research effort carried out in this study is necessary and useful as the number of studies related to experimental and dosimetry bioelectromagnetic is continuously rising. The authors experience in the field is sounding, and their comments provide important starting points for researchers in the fields of RF bioelectromagnetic interactions.
Author Response
no response required - thanks!
also the text has been polished in a number of places